# Bimodal-Structured 0.9KNbO_3_-0.1BaTiO_3_ Solid Solutions with Highly Enhanced Electrocaloric Effect at Room Temperature

**DOI:** 10.3390/nano12152674

**Published:** 2022-08-04

**Authors:** Hongfang Zhang, Liqiang Liu, Ju Gao, K. W. Kwok, Sheng-Guo Lu, Ling-Bing Kong, Biaolin Peng, Fang Hou

**Affiliations:** 1School of Physical Science and Technology, Suzhou University of Science and Technology, Suzhou 215009, China; 2Center for Advanced Ceramics, School of Materials Science and Engineering, Anhui Polytechnic University, Wuhu 241000, China; 3School of Optoelectronic Engineering, Zaozhuang University, Zaozhuang 277160, China; 4Department of Applied Physics, The Hong Kong Polytechnic University, Hung Hom, Kowloon, Hong Kong; 5Guangdong Provincial Research Center on Smart Materials and Energy Conversion Devices, Guangdong Provincial Key Laboratory of Functional Soft Condensed Matter, School of Materials and Energy, Guangdong University of Technology, Guangzhou 510006, China; 6College of New Materials and New Energies, Shenzhen Technology University, Shenzhen 518118, China; 7School of Advanced Materials and Nanotechnology, Xidian University, Xi’an 710126, China

**Keywords:** electrocaloric effect, abnormal grain growth, bimodal structure, grain-size distribution, polar nanodomain regions

## Abstract

0.9KNbO_3_-0.1BaTiO_3_ ceramics, with a bimodal grain size distribution and typical tetragonal perovskite structure at room temperature, were prepared by using an induced abnormal grain growth (IAGG) method at a relatively low sintering temperature. In this bimodal grain size distribution structure, the extra-large grains (~10–50 μm) were evolved from the micron-sized filler powders, and the fine grains (~0.05–0.35 μm) were derived from the sol precursor matrix. The 0.9KNbO_3_-0.1BaTiO_3_ ceramics exhibit relaxor-like behavior with a diffused phase transition near room temperature, as confirmed by the presence of the polar nanodomain regions revealed through high resolution transmission electron microscope analyses. A large room-temperature electrocaloric effect (ECE) was observed, with an adiabatic temperature drop (Δ*T*) of 1.5 K, an isothermal entropy change (Δ*S*) of 2.48 J·kg^−1^·K^−1^, and high ECE strengths of |Δ*T/*Δ*E*| = 1.50 × 10^−6^ K·m·V^−1^ and Δ*S/*Δ*E* = 2.48 × 10^−6^ J·m·kg^−1^·K^−1^·V^−1^ (directly measured at ***E*** = 1.0 MV·m^−1^). These greatly enhanced ECEs demonstrate that our simple IAGG method is highly appreciated for synthesizing high-performance electrocaloric materials for efficient cooling devices.

## 1. Introduction

The electrocaloric (EC) effect refers to the adiabatic temperature change in a polar material at an electric field, due to the isothermal entropy change associated with the electric-field-induced change in polarization [1,2,3]. The EC effect of ferroelectric materials has attracted continuous attention because of the potential applications in solid-state refrigeration, which is regarded as the most promising solution for cooling microelectronic devices due to the ease of miniaturization, high efficiency and low cost. Based on these considerations, a high EC performance material should possess a large isothermal entropy change (Δ*S*) and hence large adiabatic temperature change (Δ*T*) under a reasonable electric field (*E*). In other words, large EC strengths (defined by |Δ*T*/Δ*E*| and Δ*S*/Δ*E*, where parameters *T*, *S* and *E* are the temperature, isothermal entropy and applied electric field, respectively) are favored. Additionally, a wide working temperature range near room temperature (RT) is favored in order to develop high performance EC cooling devices [4,5,6]. Therefore, one critical question here is how to design and develop high-performance dielectric materials which are capable of generating giant EC effect over a broad *T*-range near RT, at a relatively low electric field.

It has been well accepted that perovskite relaxor ferroelectrics with a first-order phase transitions could be suitable solid-state systems because they present large EC effects with a mild temperature dependency and low hysteresis loss [7,8]. Moreover, owing to their even higher EC effect, bulk ceramics are superior to any type of materials, which could be implemented in medium- and large-scale cooling devices with high refrigeration capacities [9]. However, although a series of ferroelectric perovskite materials have been explored in terms of enhanced EC effect, the Δ*T* is limited, and the largest Δ*T* is less than 2 K at a relatively large electric field, corresponding to Δ*E* ˂ 50 kV·cm^−1^. This Δ*T* value does not meet the requirement of commercial applications [5]. 

Additionally, in analogy to the current trends in piezoelectric technology, highly appreciated EC materials should also be lead-free and in future be able to replace the lead-containing materials for environmental considerations. Among various ferroelectric perovskites, KNbO_3_ and BaTiO_3_, typically environmentally friendly materials, with first-order ferroelectric transitions, have been paid much attention, due to their promisingly electrical properties [10]. KNbO_3_ is a well-known ferroelectric material that exhibits the same symmetries and phase transition sequence as BaTiO_3_, with the structural phase transitions from cubic to tetragonal at a Curie temperature of *T_c_* (~435 °C), and from tetragonal to orthorhombic at ~225 °C upon cooling from high temperature to room temperature [11]. Although KNbO_3_ was theoretically calculated such that a large EC temperature change can be achieved near the Curie point, e.g., Δ*T* ~6 K at ~435 °C, while Δ*T* ~1.5 K at 120 °C for BaTiO_3_ [12], they are not desired for practical cooling device applications, due to their working temperatures being far above RT. 

Usually, the isothermal entropy change of a ferroelectric material reaches a maximum around *T_c_*, thus giving the largest EC effect. Therefore, a ferroelectric EC material is required to have a Curie temperature close to the device operation temperature, e.g., RT for most practical applications. Furthermore, the thermal hysteresis in terms of ferroelectric polarization over the phase transition region should be as weak as possible, since such a hysteresis loop represents the energy loss during the device operation [4]. Therefore, a high-quality EC system should have a Curie temperature around RT and exhibit weak thermal hysteresis. 

An immediate choice to fit the above requirement is the KNbO_3_-BaTiO_3_ solid solution which allows a remarkable reduction in the Curie temperature with respect to either KNbO_3_ or BaTiO_3_. For example, the 0.9KNbO_3_-0.1BaTiO_3_ solid solution, denoted as KN-BT(9/1), exhibits a Curie temperature *T_c_* of ~50 °C, which is very close to RT. In addition, the two distinctly different *T_c_* of the ferroelectric components favor the formation of a diffused phase transition between the high-temperature paraelectric phase and the low-temperature ferroelectric phase, and thus a weak thermal hysteresis [11]. 

Nevertheless, the EC effect of KN-BT(9/1) has been rarely reported, mainly because the preparation of highly dense KN-BT(9/1) ceramics using the conventional solid-state reaction technology is still a challenge due to the large differences between the properties of the cations in KNbO_3_ and BaTiO_3_. The melting temperature of KNbO_3_ is also far lower than that of BaTiO_3_ (1040 °C vs. 1625 °C). Moreover, the high volatility of potassium oxide (K_2_O) at high temperature, such as 800 °C, will result in non-stoichiometry and thus the formation of unwanted phases [13]. A novel sintering technique to prepare high quality KN-BT(9/1) ceramics is thus highly appreciated. 

In this study, the newly developed sintering technique, the induced abnormal grain growth (IAGG) method [14], is applied to prepare KN-BT(9/1) ceramics with a novel bimodal grain size distribution in a simple and easy way. The bimodal grain size distribution structure (briefly called the bimodal structure) consists of a small fraction of large-sized grains uniformly embedded in the matrix of fine grains. The large grains are evolved from the micron-sized KN-BT(9/1) powders (filler), while the small ones are derived from the nano-sized KN-BT(9/1) powders (matrix). The highly dense and stoichiometric KN-BT(9/1) ceramics are well sintered at a relatively low sintering temperature, and more importantly, exhibit a strong EC effect near RT at a low electric field. 

## 2. Materials and Methods 

### 2.1. Preparation of the Filler and Matrix Powders 

Micron-sized KN-BT(9/1) filler powder was fabricated using KNbO_3_ and BaTiO_3_ as raw powders by using the conventional ceramic processing. The KNbO_3_ powder was prepared from pure grade K_2_CO_3_ (99.99%, Aladdin) and Nb_2_O_5_ (99.95%, Alfa Aesar) powders, which were first ball-milled and calcined at 640 °C for 4 h. Then, the calcined powders were ball-milled again for 24 h. After drying at 120 °C overnight, the KNbO_3_ powders were mixed thoroughly with a commercial-grade high purity nano-sized BaTiO_3_ powder (D_50_ = 50 nm, 99.9 wt.% purity, SAKAI Chemical Industry Co. Ltd., Osaka, Japan), in a molar ratio of 9:1, by ball-milling. The mixture was then calcined at 900 °C for 2 h, ball-milled again for 24 h, and dried at 120 °C overnight to produce the KN-BT(9/1) filler powder (Appendix A).

A modified Pechini method was introduced to prepare KN-BT(9/1) sol precursor. All raw materials were weighed according to the designed composition. For Nb-sources, Nb_2_O_5_ powder was dissolved in hydrofluoric acid (48–51%, ACS, Alfa Aesar) at 80 °C. Then, ammonium hydroxide (28% NH_3_, Alfa Aesar) was tardily added into the solution until the pH value reached 10, followed by filtering, washing and drying of the sediment at 80 °C for 10 h. Subsequently, niobium hydroxide was formed. The Nb source was then obtained by dissolving the niobium hydroxide in a citric acid (CA) solution. For preparing the metal–CA solution (metal: K and Ba), K_2_CO_3_ and BaCO_3_ (99.95%, Aladdin) were dissolved directly in a CA solution. For the Ti–CA solution, tetrabutyltitanate (C_16_H_36_O_4_Ti, 96%, Alfa Aesar) with the CA solution was heated to 80 °C until the solution became transparent. Finally, all the metal sources were mixed, with a molar ratio of CA:EG (ethylene glycol) to be 1/4 and pH value to be 10. KN-BT(9/1) precursor solution was obtained by stirring the solution at 80 °C for 2 h. The precipitant was dried at 120 °C for 24 h and then calcined at 900 °C for 2 h as a matrix powder. The detailed synthesis routes are schematically shown in Appendix A. 

### 2.2. Preparation of Bimodal Structured KN-BT(9/1) Ceramics 

The IAGG method is schematically shown in Appendix A. The micron-sized KN-BT(9/1) filler powder and the nano-sized KN-BT(9/1) matrix powder were first mixed thoroughly by ball balling in ethanol for 4 h. After drying at 120 °C, the mixture was then uniaxially pressed into green pellets with a diameter of 10 mm and a thickness of about 1 mm. Finally, the green pellets were sintered at 1000–1050 °C for 2 h in air with a heating rate of 2 °C·min^−1^. For comparison, the filler powder and the matrix powder were also used separately to prepare KN-BT(9/1) ceramics by similar procedures. All the samples were cooled by natural cooling in the furnace.

### 2.3. Characterization

#### 2.3.1. Microstructure and Morphology 

Crystal structures were examined using X-ray diffraction (XRD, D8 ADVANCE, Bruker AXS Corporation, Karlsruhe, Germany) with CuK*_α_*_1_ radiation (1.5418 Å, 40 kV, 40 mA) from 2 theta (degree) = 15 to 70° at a scan step of 0.02° per second at room temperature. The Rietveld refinement of XRD pattern was carried out using the GSAS-EXPGUI software package (General Structure Analysis System, A.C. Larson and R.B. Von Dreele, Los Alamos National Laboratory Report LAUR 86-748 (2004); B.H. Toby, EXPGUI, a graphical user interface for GSAD, J. Appl. Cryst. 34, 210-21 (2001)). Microstructures of the ceramic samples were observed by using scanning electron microscopy (SEM), equipped with energy dispersive X-ray spectroscopy (EDS) (JSM-6335F, JEOL Japan Electronics Co., Ltd., Kyoto, Japan) at 30 kV. The morphology, microstructures and polar nanodomain regions (PNRs) were observed using a high-resolution transmission electron microscope (HRTEM, Tecnai G2 F20 S-Twin, FEI, Hillsboro, OR, USA), acceleration voltages 200 kV, spot size 2.

#### 2.3.2. Measurement of Specific Heat Capacity

The specific heat capacity of the samples was measured by using the Mettler Toledo DSC3 instrument according to the Sapphire method. The heat flow was measured directly in the temperature range of −50–200 °C such that the specific heat capacity can be given by
(1)$$C_{p,sample}=\frac{\Phi_{sam}}{\Phi_{sap}}\frac{m_{sap}}{m_{sam}}C_{p,sap}$$
where *C_p,sam_* and *C_p,sap_*, *Φ_p,sam_* and *Φ_p,sap_*, *m_sam_* and *m_sap_*, are the specific heat (J/K·g), heat flow (W·g^−1^), and mass of the sintered KN-BT(9/1) bulk sample at 1050 °C and standard sapphire as reference, respectively.

#### 2.3.3. Characterization of Dielectric and Ferroelectric Properties

To measure dielectric and ferroelectric properties of the ceramics, two sides of the disc samples were coated with Ag paste fired at 600 °C for 30 min as electrodes. Temperature-dependent dielectric characteristics were measured over 1–100 kHz by using a dielectric analyzer (TZDM-RT-800, Harbin Julang Technology Co., Ltd., Harbin, China) over −20–500 °C, at a rate of 1 K·min^−1^. Ferroelectric hysteresis (P-E) loops were recorded by using a modified Sawyer–Tower circuit method operated at a frequency of 10 Hz, over the temperature range from RT to 52 °C using the power supply Trek Model 610C.

#### 2.3.4. Measurement of Electrocaloric Effect

Direct ECE measurement was carried out in this study. To directly measure the ECE signals, a pulsed electric field was applied to the sample with a thermocouple (Precision Fine Wire Thermocouple, Omega Engineering, Inc., Norwalk, CT, USA) attached directly to record the temperature variation as shown in Figure 1.

For measurement, the sample was hung in air through two wires to avoid any heat dissipations. The thermocouple that directly touched one surface of the sample (ground side) was connected to an oscilloscope (Teledyne LeCroy WaveSurfer 3024 Oscilloscope 200 MHz) to record the temperature, power supply: Trek Model 610C. The measurement was carried out by (i) manually applying an electric field to the sample with a positive peak appearing on the oscilloscope, (ii) waiting for the heat peak to completely pass and temperature curve to become constant, then manually removing the electric field, and (iii) showing a cooling peak. No constant pulse wide of the electric field was set. Based on the dimension of the samples, the period was about 2 s. The final cooling performance was obtained by calculating the temperature difference between the initial temperature and maximum value of the cooling peak. The precision and validity of thermocouple were checked as follows. Firstly, a heat plate with a known temperature, which was confirmed by using another infrared (IR) thermometer, was used to check the precision of the thermocouple, where the temperature read from the thermocouple should be in perfect agreement with that read from the IR thermometer. Secondly, before the ECE of the ceramic samples was calculated, a Teflon plate with the same Au electrode coated on each side was used to check the validity. As shown in the schematic diagram, a Teflon plate was used to replace the ceramic samples, and a similar electric field was applied. No temperature changes can be read from the thermocouple. Therefore, the observed temperature changes should be the real temperature changes of ceramics.

## 3. Results and Discussion

### 3.1. Phase Composition and Microstructure of the Bimodal Structured Ceramics

Figure 2a shows XRD patterns of the commercial-grade nano-sized BaTiO_3_ powder, micron-sized KNbO_3_ powder (calcined at 640 °C), the micron-sized KN-BT(9/1) filler powder and the nano-sized KN-BT(9/1) matrix powder calcined at 900 °C respectively, together with the bimodal structured KN-BT(9/1) ceramics sintered at 1050 °C using the IAGG method. The TEM (transmission electron microscope) images of the filler and matrix of KN-BT(9/1) powders calcined at 900 °C are shown in Figure 2b,c.

As shown in Figure 2a, all the samples exhibit a typical perovskite structure, while the commercial nano-sized BaTiO_3_ powder exhibits a cubic perovskite structure (PDF: 31-174) [15]. As is well known, for perovskite solid solutions, the orthorhombic and tetragonal structures experience different lattice distortions with respect to the cubic structure. The orthorhombic phase (O-phase) and tetragonal phase (T-phase) can be identified according to the peak splitting, with (220)_O_/(002)_O_ and (200)_T_/(002)_T_ peaks for the O-phase and T-phase, respectively. Therefore, the calcined KNbO_3_ powder exhibits the O-phase due to the (220)_O_/(002)_O_ splitting with higher left peak than the right one (PDF: 32-0822) [16]. On the other hand, the (IAGG-prepared) ceramic sintered at 1050 °C possesses a tetragonal phase, as evidenced by the splitting of the diffraction peaks (200)/(002) located near 44–47° (inset of Appendix A), which is in agreement with the phase transition diagram of the KN-BT system [11]. Moreover, the Rietveld refinement of the XRD pattern is shown in Appendix A, including Appendix A showing the corresponding crystal structure information using the Rietveld method respectively. By comparison, Appendix A shows the Rietveld refinement of the parent XRD patterns, i.e., micron-sized pure KNbO_3_ powders calcined at 640 °C, and commercial-grade nano-sized pure BaTiO_3_ powders, including the corresponding Rietveld refinement information in Appendix A respectively.

As observed in Figure 2b,c, the micron-sized KN-BT(9/1) powder, as the filler, consists of the irregular crystalline particles with grain sizes of about 200–500 nm, while the nano-sized KN-BT(9/1) powders, as the matrix, are characterized by uniform rectangle crystalline particles with an average grain size of about 100 nm.

Additionally, Appendix A shows XRD patterns of the KN-BT(9/1) bulk ceramics prepared by using the conventional ceramic processing (Appendix A), the sol-gel technique (Appendix A) and IAGG method sintering at 1050 °C. All the samples exhibit the single-phase perovskite structure.

Figure 3 shows the SEM images and grain size distribution profiles of the KN-BT(9/1) ceramics fabricated using the IAGG method. As shown in Figure 3c,d, the average grain size and size distribution of the bimodal structure of the KN-BT(9/1) bulk ceramics sintered at 1000 and 1050 °C were evaluated by using the software equipped with the SEM equipment with both surface and cross-sectional SEM images at different magnifications. Clearly, the bimodal structure is demonstrated by the two well-separated distribution peaks. For the sample sintered at 1000 °C, only a small number of grains are larger than 1.0 μm in size, surrounded by nano-sized grains of ~0.1 μm. When the sintering temperature was increased to 1050 °C, the relatively large grains began to be exaggerated and elongated. Eventually, ultra-large grains with sizes of ~10–50 μm were formed. Therefore, our IAGG method is an effective way to develop bimodal grain-size distribution, with a small number of coarse grains uniformly distributed in the fine-grained matrix.

Here, it is very important to ensure the composition homogeneity in the coarse and fine grains. To demonstrate this, Figure 3e shows the cross-sectional surface image with high amplification. The EDS spectra of the coarse and fine grains are plotted in Figure 3f, and the automated element identification for the EDS spectra evaluation is shown in Appendix A respectively. Indeed, the stoichiometry of the large grains is very close to that of the fine-sized grains within the measuring uncertainties. Therefore, it is confirmed that the IAGG method is powerful for obtaining such a bimodal structure while maintaining composition homogeneity over the whole samples. In other words, the IAGG method did not trigger the uneven distribution of K^+^ and Nb^5+^ in the two types of grains.

Figure 4 shows HRTEM images and the corresponding SAED (selected area electron diffraction) patterns of the coarse and fine grains. As shown in Figure 4b,c,e,f, on the one hand, the almost identical lattice planes demonstrated homogenous structure of the samples. On the other hand, the indexed KN-BT(9/1) grains exhibit the characteristic T-phase structure (PDF #71-0945). Therefore, it is concluded that the bimodal structured KN-BT(9/1) bulk ceramics have a typical T-phase as mentioned above.

For comparison, surface morphologies of the KN-BT(9/1) bulk ceramics fabricated using the conventional ceramic processing and sol-gel technique are shown in Appendix A, together with the corresponding grain size distribution profiles (c,d). As expected, these two KN-BT(9/1) bulk samples with unimodal structures, with grain sizes of about 250 nm and 300–400 nm, respectively. No abnormal grain growth (AGG) phenomena are observed in the samples sintered at 1050 °C.

Grain growth behavior of the KN-BT(9/1) ceramics prepared by using the IAGG method can be understood with the explanation of Kingery and Kanget et al. [17,18]. Due to the difference in free-energy across a curved grain boundary, the irregular micron-sized KN-BT(9/1) filler powders with large curvature, underwent exaggerated growth, acting as a “seed” to consume the neighboring nano-sized ones in the matrix. Therefore, in a given polycrystalline system, the grain growth behavior is governed by the maximum driving force (Δ*g_max_*) relative to the critical driving force (Δ*g_c_*), showing the mixed controlling growth behavior. Although the grain size could be increased by sintering at very high temperatures or for very longer times in theory, no grain growth with specific morphologies and crystal orientations of large grains occur when Δ*g_max_* is smaller than Δ*g_c_*. Therefore, our IAGG method in this study is an effective way to develop ceramics with extraordinarily oriented large grains at relatively low sintering temperatures when using the solid-state reaction process. It seems that the filler grains are “cloned”, while the gel matrix is just like a “nutrient source or reservoir” to breed the fillers to grow. This method is simple, reproducible and low cost, which can be easily extended many other ferroelectric perovskite materials.

### 3.2. Dielectric and Ferroelectric Properties

Figure 5 shows dielectric properties dependent of the temperature (*ε_r_* (*T*)) at different frequencies, and the microstructure of the KN-BT(9/1) bulk ceramics sintered at 1050 °C. As shown in Figure 5a, all the *ε_r_* (*T*) curves exhibit a broad peak centered at about −20~100 °C, indicating that diffused phase transition is present in the KN-BT(9/1) bulk ceramic [11,19]. At RT, the relative permittivity ($\varepsilon_{r}$) and loss tangent are about 792 and 0.067 at 1 kHz, respectively. The diffused behavior is further confirmed by the dielectric loss curves.

For relaxor ferroelectrics, the reciprocal of relative permittivity as a function of temperature, follows the Uchino and Nomura function, a modified Curie–Weiss law, which is expressed as [20]
(2)$$\frac{1}{\varepsilon_{r}}-\frac{1}{\varepsilon_{m}}=\frac{{\left(T-T_{m}\right)}^{\gamma }}{C}$$
where C is the Curie constant and γ is the diffusion coefficient ranging from 1 (an ideal normal ferroelectric) to 2 (an ideal relaxor ferroelectric); $\varepsilon_{m}$ and $T_{m}$ are the maximum relative permittivity and corresponding temperature at a fixed frequency, respectively. The slope of the fitting curves is used to determine the γ value in the Figure 5b. The value is γ = 1.59 at 100 kHz, confirming the relaxor-like ferroelectric behavior of the KN-BT(9/1) bulk sample.

The ferroelectric nanodomains were observed through HRTEM images, as shown in Figure 5c, displaying grains with sizes of 2–10 nm randomly in the nondomain matrix. The presence of the PNRs provides strong evidence of the diffusion phase during the phase transition in the KN-BT(9/1) bulk sample. The diffused phase transition can be ascribed to the partial breaking of the ferroelectric long-range ordering by the coupled substitutions of Ba^2+^ and Ti^4+^ ions for K^+^ and Nb^5+^ ions with different sizes and charges, respectively, due to the simultaneous occupation of the six-coordination site by Ti^4+^ and Nb^5+^ [11,21,22].

Moreover, it is believed that the presence of the fine-size grains in the bimodal structured sample is responsible for additional decrease in the $T_{m}$ [23].

As illustrated in Figure 5d, no obvious anomaly is observed in the specific heat capacity. The specific heat capacity value is increased with increasing temperature, with a room temperature value of 0.50 J·K^−1^·g^−1^. As shown in Appendix A, a weak anomaly is present on the heat capacity curve, which is similar to the observation of Pb-free relaxor Ba(Ti_0.65_Zr_0.35_)O_3_ ceramics [24].

Figure 6a shows P-E hysteresis loops of the KN-BT(9/1) bulk sample sintered at 1050 °C measured at RT and 52 °C at 10 Hz. Figure 6b illustrates P-E curves measured at different electrical fields, while the curve of the remanent polarization versus the electric field at RT is shown as the inset in Figure 6b.

The sample has a ferroelectric nature, whereas the slim P-E hysteresis loops suggest that the ceramics have low hysteresis losses. The P-E loops at RT and 52 °C were nearly the same, indicating that the presence of ferroelectricity in nature can be retained over a relatively broad temperature range. Additionally, as seen in Figure 6b, the P-E loop is electric field dependent, with the remanent polarization to be increased almost linearly with the increasing electric field. At 2.5 MV·m^−1^ and RT, the values of the *P*r and the coercive field (*E*_C_) are 0.675 μC·cm^−2^ and 0.23 MV·m^−1^, respectively. By comparison, the P-E loops of the KN-BT(9/1) bulk samples sintered at 1050 °C using the conventional ceramic processing and sol-gel technique are shown in Appendix A. Obviously, lossy hysteresis loops are present, indicating higher conductive behavior in a unimodal structure. Therefore, our bimodal structured bulk ceramic displays a relatively good ferroelectric property. It can be inferred that the extra-large grains ensure the ferroelectricity, while the fine grains ensure high density to suppress the tunneling current [25].

### 3.3. Electrocaloric Effect

ECE adiabatic ∆T refers to the temperature drop induced after removing the electric field. The typical thickness of the KN-BT(9/1) bulk sample used in the ECE measurement was 0.264 mm with an *electrode diameter* of 6 mm. The specific isothermal entropy change, Δ*S*, is calculated with Δ*S* = *c* ΔT/T, where *c* is the specific heat of the ceramic sample [26]. Figure 7a shows the directly recorded ECE signal of the bulk sample at RT at 1 MV·m^−1^, where the temperature is demonstrated to rise and drop as the field is applied and removed. The Δ*T* and Δ*S* at RT at different electric fields are presented in Figure 7b. The ratios of |ΔTΔE| and ΔS/ΔE (or ΔQ/ΔE, where ΔQ=TΔS) are used to express the electrocaloric coefficients (ECE strengths). It is found that the KN-BT(9/1) ceramics have high ECE at ***E*** = 1 MV·m^−1^, corresponding to Δ*T* = −1.5 K and Δ*S* = 2.48 J·kg^−1^·K^−1^. Accordingly, |ΔTΔE| = 1.50 × 10^−6^ K·m·V^−1^ and ΔS/ΔE = 2.48 × 10^−6^ J·m·kg^−1^·K^−1^·V^−1^ were obtained at RT. As discussed above, the strongly widened phase transition temperature near RT is responsible for the giant ECE response over a relatively broad temperature range [4,5].

Additionally, as shown in Table 1, compared with the ferroelectric ceramics reported in refs. [8,9,25,26,27,28,29,30,31], our bimodal structured KN-BT(9/1) bulk ceramics shows a fairly high ECE coefficient (strength), which is close to the ECE of single crystal BaTiO_3_ at 10 °C. Besides the diffused phase transition temperature near RT, the coarse grains (10–50 µm) in the bimodal structure should enhance the dielectric and ferroelectric properties of the sample, resulting in high ECE strength at relatively low electric fields [23]. At the same time, the finer grains derived from the matrix play a crucial role in forming a dense microstructure and inhomogeneous dielectric properties, leading to high entropy [3].

However, the unimodal structured KN-BT(9/1) made by using the conventional ceramic processing and sol-gel technique cannot be used to measure the ECEs, apparently, the novel bimodal structured KN-BT(9/1) bulk ceramics by the IAGG method can overcome the shortage of the unimodal structured samples. Additionally, the preparation of the bimodal structured KN-BT(9/1) ceramics using the IAGG method is highly compatible with the conventional ceramic process, giving them potential as micro-refrigerators to be used for cooling the microelectronic devices near RT.

## 4. Conclusions

Bimodal structured KN-BT(9/1) bulk ceramics with a tetragonal phase at RT and a diffused phase transition were prepared successfully by using the IAGG method at a relatively low sintering temperature of 1050 °C. In this bimodal structure, the exaggeratedly large grains were evolved from the micron-sized KN-BT(9/1) filler powders, while the fine grains were originated from the KN-BT(9/1) sol precursor matrix. As compared with the unimodal structured counterpart, the bimodal structured KN-BT(9/1) bulk ceramics display a high electrocaloric performance, giving a large ECE-induced adiabatic temperature drop of 1.5 K and a large EC coefficient of 2.48 × 10^−6^ J·m·kg^−1^·K^−1^·V^−1^ at RT, which is advantageous to the design of cooling devices. It is believed that the coarse grains engender the high ferroelectricity and ECE strengths, while the fine grains are responsible for the decreased maximum temperature, and the enhanced density. Our IAGG method is simple, reproducible and cost effective, which can be easily extended to other ferroelectric perovskite materials.

## Figures and Tables

**Figure 1 nanomaterials-12-02674-f001:**
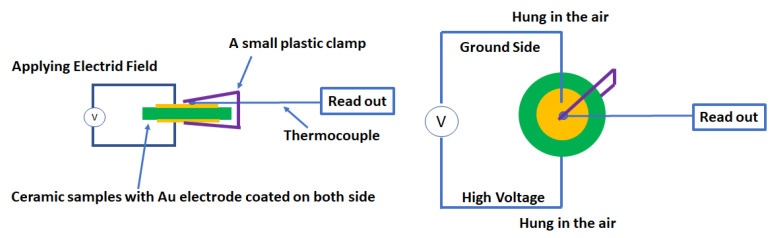
Schematic diagram of the electrocaloric effect measurement setup.

**Figure 2 nanomaterials-12-02674-f002:**
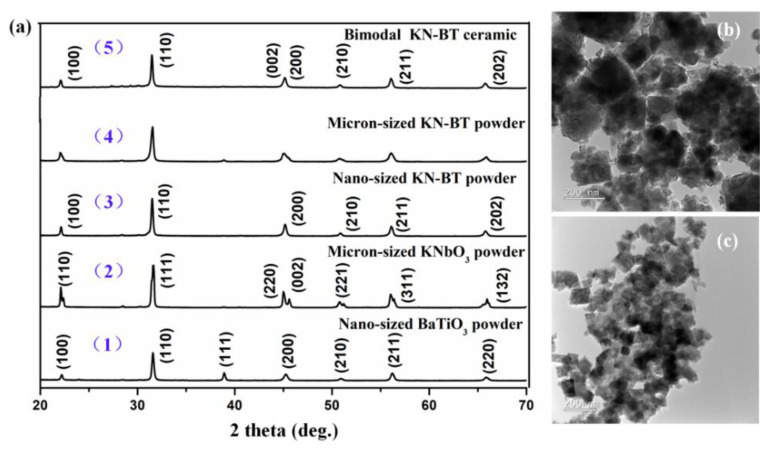
(**a**) XRD patterns of the (1–4) powders: (1) the commercial nano-sized BaTiO_3_, (2) micron-sized KNbO_3_, (3) nano-sized KN-BT(9/1) matrix powders, and (4) micron-sized KN-BT(9/1) filler powders, including (5) XRD pattern of the bimodal structured KN-BT(9/1) bulk ceramic. (**b**,**c**) TEM images of the micron-sized filler and nano-sized matrix powders calcined at 900 °C respectively.

**Figure 3 nanomaterials-12-02674-f003:**
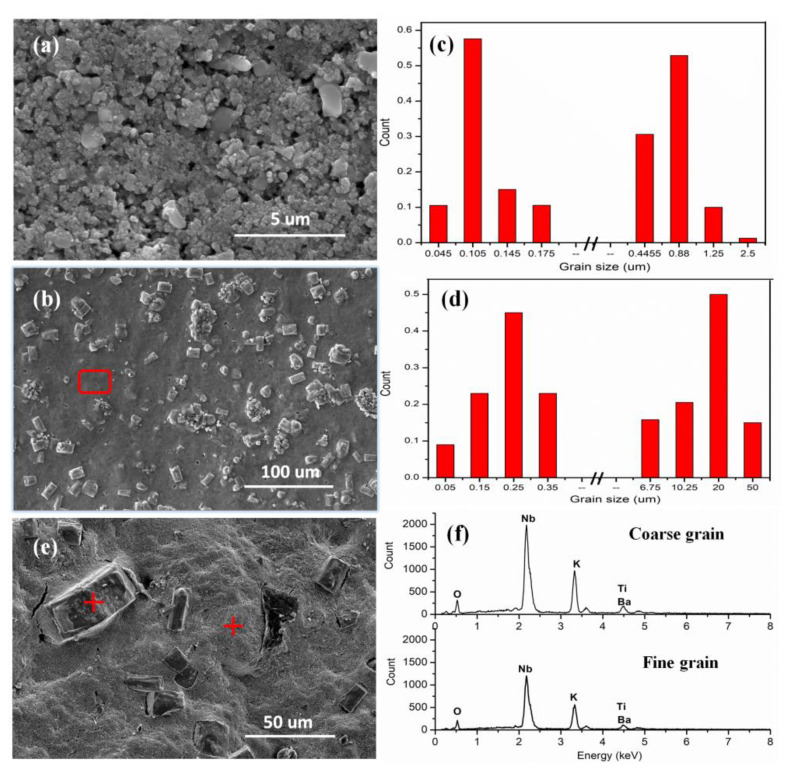
SEM images of the KN-BT(9/1) bulk ceramics sintered at (**a**) 1000 °C and (**b**) 1050 °C, together with (**c**,**d**) corresponding grain size distribution profiles. (**e**) Cross-sectional SEM image of the KN-BT(9/1) bulk ceramic sintered at 1050 °C, with (**f**) its corresponding EDS results, marks indicating the selected coarse and fine grains.

**Figure 4 nanomaterials-12-02674-f004:**
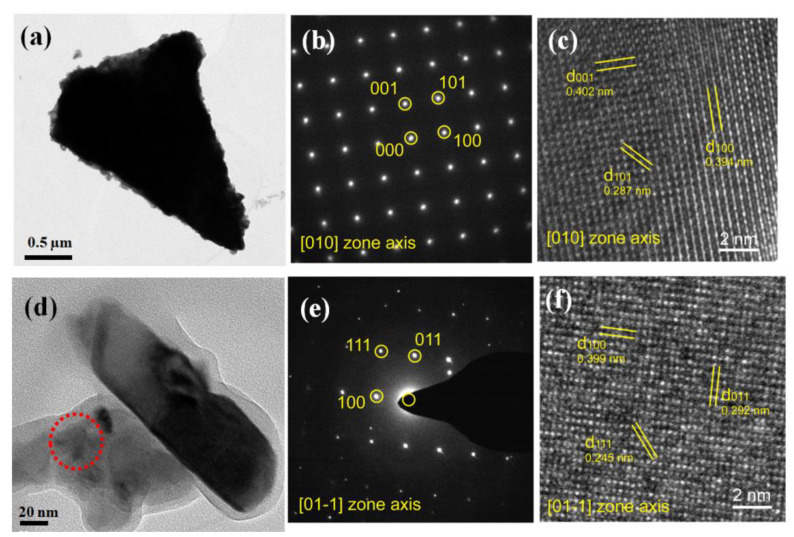
HRTEM images and SAED patterns of the coarse grain (**a**–**c**) and small grain (**d**–**f**). The bimodal structured KN-BT(9/1) bulk ceramic sintered at 1050 °C was selected for the analysis. Notes: the lattice planes 001, 101, 100 in [010] zone axis respectively in (**b**); as well as the lattice planes 111, 011, 100 in [01-1] zone axis respectively in (**e**).

**Figure 5 nanomaterials-12-02674-f005:**
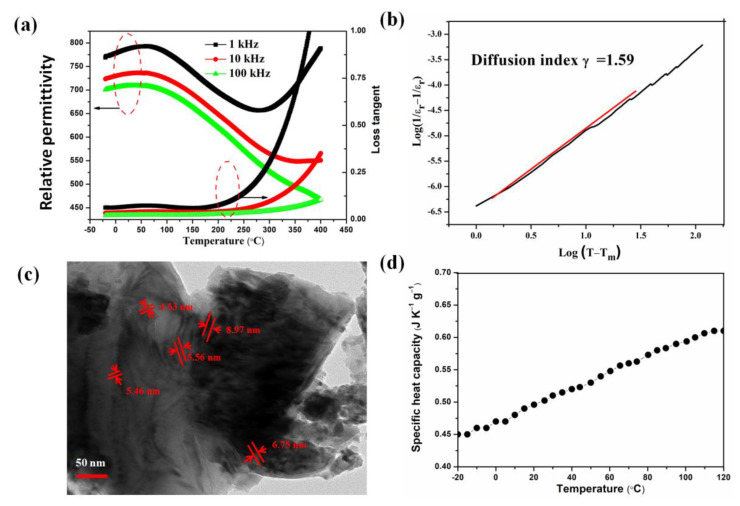
(**a**) Dielectric properties versus temperature at three selected frequencies (1 kHz, 10 kHz and 100 kHz). (**b**) Plot of log($\frac{1}{\varepsilon_{r}}-\frac{1}{\varepsilon_{m}}$) as a function of log$\left(T-T_{m}\right)$ at 100 kHz. (**c**) TEM image of polar nanodomain regions (PNRs) in the sample. (**d**) Measured specific heat capacity of the KN-BT(9/1) ceramics. Note that the bimodal structured KN-BT(9/1) bulk ceramic sintered at 1050 °C using IAGG method was selected for the analysis.

**Figure 6 nanomaterials-12-02674-f006:**
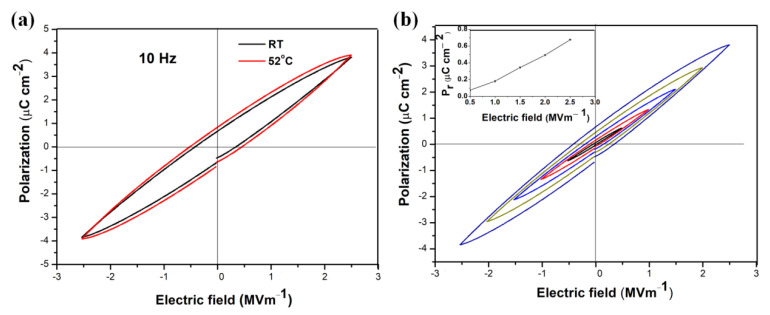
(**a**) P-E loops measured at RT and 52 °C at 10 Hz. (**b**) P-E loops measured at different electric fields at RT, with the inset showing the remanent polarization (*P*r) versus electric field. Note that the bimodal structured KN-BT(9/1) bulk ceramic sintered at 1050 °C using the IAGG method was selected for the analysis.

**Figure 7 nanomaterials-12-02674-f007:**
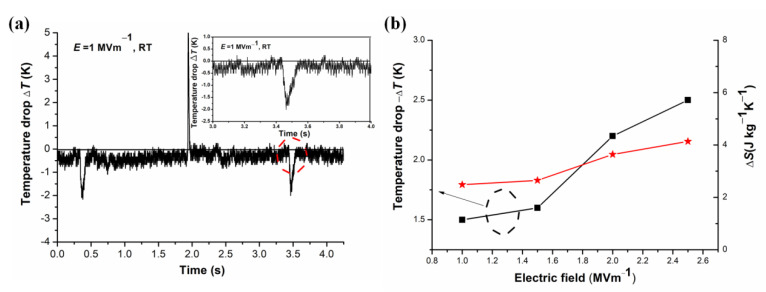
(**a**) Directly recorded ECE signal for the KN-BT(9/1) sample as the electric field was switched on and off. Solid line is drawn to show the ambient temperature. The amplified temperature drop is shown as the inset. (**b**) ECE–induced adiabatic temperature drop (ΔT) and isothermal entropy change (ΔS) as a function of electric field (***E***) at RT. The dash line is drawn to guide eyes. Note that the bimodal structured KN-BT(9/1) bulk ceramic sintered at 1050 °C using IAGG method was selected for the analysis. Error: within the noise range, it is about ± 0.2 °C.

**Table 1 nanomaterials-12-02674-t001:** ECEs of different ferroelectric ceramics with operating temperature near RT.

Material	Form	*T* (°C)	ΔT (K)	*E* (MV m^−1^)	|ΔTΔE| (10^−6^ Km V^−1^)	ΔS/ΔE (10^−6^ Jm kg^−1^ K^−1^ V^−1^)	Method	Reference
KN-BT(9/1)	Ceramic	23	1.5	1	1.50	2.48	Direct	This work
Ba(ZrxTi_1–x_)O_3_	Ceramic	38	1.1	2.1	0.52	0.93	Direct	[26]
Bi_0.5_Na_0.5_TiO_3_-based	Ceramic	23	0.45	3	-	-	Direct	[9]
Ba_0.65_Sr_0.35_TiO_3_	Ceramic	23	0.4	2	0.21	-	Indirect	[27]
(Pb, La_,_ Ba)(Zr, Sn, Ti)O_3_	Ceramic	30	0.25	2.2	0.11	-	Direct	[28]
PZT-5	Ceramic	30	0.15	2.8	0.05	-	Direct	[29]
(Ba, Ca)(Zr, Ti)O_3_	Ceramic	60	0.3	2.0	0.15	-	Indirect	[30]
0.9PMN-0.1PT	Ceramic	25	0.63	2.8	0.23	-	Direct	[25]
BaTiO_3_	Single crystal	10	1.4	1	1.4	-	Direct	[31]
0.9PMN-0.1PT	Single crystal	50	1.0	4.00	0.25	-	Indirect	[8]

## Data Availability

The data presented in this study are available in this article and Appendix A.

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
