# Peer review of "Bimodal-Structured 0.9KNbO3-0.1BaTiO3 Solid Solutions with Highly Enhanced Electrocaloric Effect at Room Temperature"

_nanomaterials, 2022, doi:10.3390/nano12152674_

Round 1
Reviewer 1 Report
The manuscript is interesting especially because of its applicative aspect. However, the text of the manuscript is very difficult to read due to the large number of abbreviations used by the authors. The abbreviations used in the text refer not only to phenomena but also to chemical formulas. Although the authors explain the meaning of abbreviations, such a large number of them causes the reader to perceive such a situation as a haos. I suggest authors to use at least chemical formulas instead of abbreviations.
Reviewer 2 Report
This manuscript reports on high electrocaloric (EC) effect obtained at room temperature on 0.9KNbO3-0.1BaTiO3 ceramics prepared by an induced abnormal grain growth (IAGG) method. The reported research is quite interesting but the manuscript message is rather hard to follow due to disadvantages in both the manuscript language and structure besides the lack of many details and some mistakes and contradictions.
The language difficulty is seen right from the title that should be rephrased. Then in the abstract (and in general in manuscript) it is enhanced by absence of use of sub- and superscripted symbols when needed, of decoding for abbreviations like HRTEM and ECE, of some Greek symbols and of unit stile unity (when Jmkg-1K-1V-1 are used simultaneously with MV/m). Then there are also many grammar mistakes, missing words or spaces between words as well as abbreviations like FE, SEM, TEM, SAED or symbols like εr, εm, Tm and P-E not decoded at first mentioning or at all. Further difficulty is due to the fact that the figure descriptions are usually limited to one sentence like figure caption, while figure captions do not identify many details starting from Fig. 2. Moreover, incorrect calling of the dielectric permittivity as dielectric constant results in odd phrases like “dielectric constant (??,max) increases”. In addition, scientific manuscripts should avoid estimative words like “good” or “excellent” since they depend on point of view.
Concerning the manuscript structure issues, it seems that the only results reported for IAGG ceramics sintered at 1000 C are microstructure and grain size distribution in Figs. 3a and 3c. However, these not much needed results confuse the reader regarding other presented data. Thus, it is unclear ECE for 1000 or 1050 C ceramics was studied and shown in Fig. 7 as well as there are no sintering temperature indications in captions of Figs. 5 and 6.
At the same time, there are many missing details including the key ones. Indicating the transition temperatures for KN, those for BT should also be presented. Then, writing that “micron-sized KN-BT(9/1) particles was dispersed in nano-sized KN-BT(9/1) powders (as the matrix), and then ball milled for 4 h” the authors have to present a ratio of the mixed powders and show the particle size distributions allowing them to call one of the powders as micron-sized and another as nano-sized. Moreover, they have to indicate the pot and ball materials as well as rotation rate used for the milling step here and during the micron-sized KN-BT(9/1) particles preparation. In addition, they have to provide the pressure value used to prepare green pellets and the cooling rate value at the sintering step. Purity and trademarks for BaCO3, K2CO3 and Nb2O5 raw powders have also to be shown. Then, XRD range and rate as well as SEM, EDS, TEM acceleration voltages have to be indicated. Regarding ECE setup, the power supply and oscilloscope trademarks and models have to be shown as well as the estimated measurement error. The error of heat capacitance measurement has to be indicated as well.
The missing results detail are following. (200) XRD peak has to be shown and discussed not only for ceramics and KN powder but for all the powders presented in Fig. 3. Moreover, the lattice parameter values have to be deduced from XRD with particular need for presentation of relative ceramic density before the discussion of its electrical properties. It would be also nice if the authors could define the crystallite size values from their XRD data. Concerning the EDS analysis results, they have to be presented together with associated error values. Moreover, there seems to be also some Si contamination from over 24 h milling that has to be estimated. From HRTEM and SAED results, the authors conclude on T-phase structure of their ceramics, not providing however detailed discussion on the reasons to exclude O-phase and C-phase.
Showing the ionic size values, the authors have to provide a reference or a method, which they used to define them. Then they have to prove or explain in detail why “When Ti4+ was substituted by Nb5+, the shift (Δ?) of the tetravalent cation from the octahedron centeris limited”. To see the loss tangent in details in Fig. 5a the authors have plot it in log scale.
It is also unclear what the authors mean by “the diffused phase transition can be ascribed to the freezing of the displacive-type phase transition” at line 328 and by “relaxor ferroelectrics (RFE) with a similar first-order phase transitions” at line 54. How the phase transition can be frozen and that freezing can be another phase transition? How RFE with smooth temperature variation of polarisation can be simultaneously with first-order phase transitions characterised by sharp polarisation variation at the transition temperature?
Concerning the contradictions and mistakes, Ref. 12 report on temperature of 120 C but not 12 C, as written by the authors at line 76. At lines 310-312, the authors wrote “it is obvious that the temperature (??) of the maximal dielectric constant (??,max) increases with increasing frequency”. However, corresponding Fig. 5a shows opposite variation. rTi4+ = 0.605 Å but not 0.604 Å according to classical work by Shannon. A stronger mistake is related with equation ΔS = c ρ?Δ?/?, which is shown by the authors without any reference, without numbering and most important without any relation between ΔT and ΔS values reported in this work. Moreover, according to this equation ΔS units should be J/K, while units shown by the authors are J/(K kg). The units are also incorrect in ΔT/ΔE and ΔS/ΔE columns of Table 1. ΔS/ΔE value from Adv. Funct. Mater. 2014, 24, 1300 do not correspond to the one in the table, while the one attributed by the authors to AIP Adv. 2012, 2, 022162 in not mentioned in this cited work at all. Furthermore, Ref. 7 has wrong publication year, while reference numbering has a problem since the range of Refs. 16-18.
As a result, current manuscript can be considered for publication, only when all the mistakes are corrected, all the doubts are clarified, all the details are provided and the manuscripts is revised according to the comment above.
Reviewer 3 Report
Referee Report
on paper “Preparation and Highly Enhanced Electrocaloric Effect in Bi-modal-Structured 0.9KNbO3-0.1BaTiO3 Solid Solution at Room Temperatures“ (nanomaterials-1800530) to Nanomaterials
This is interesting paper. It reports about the improvement of the electrical properties of the KNbO3-BaTiO3-based composites. Authors prepared KNO-BTO composites by IAGG. It is an interesting that authors use special approaches for functionalization of the fine structure and enhancement dielectric properties. The obtained experimental results are interesting and reliable. However, paper needs some improvement only after which it can be accepted. At this stage, my decision is major revision. But I hope that after brutal revision and replies on my comments this paper can be accepted. I impressed by the paper. Hope it will be accepted.
1. Title: I propose check lower indexes in chemical formula in the title. The same recommendation for number of the affiliations in author’s list and abstract (upper indexes).
2. Abstract: I recommend to delete part of information and highlight most important result of the paper (authors can cut part with processes description). Also please use full title before abbreviation (for example HRTEM).
3. I feel that Introduction seems poor. The choice of the research object is attractive. This is complex transition metal oxide based on well known KNbO3 and BaTiO3. I also agree with authors that the complex transition metal oxides attract great attention, please discuss scientific and practical importance of the different complex oxides and their composites for energy-storage applications (https://doi.org/10.1016/j.nanoen.2020.105247, https://doi.org/10.1016/j.ceramint.2018.08.180; https://doi.org/10.3390/nano10040801, https://doi.org/10.1039/D0TA03304A,) in Introduction.
4. Please highlight the motivation of the method for sample preparation.
5. The Tab. On inserts (Fig. 3) have poor resolution. It is uninformative. I propose use larger letters and numbers. May be present this as separate tables (not as inserts).
6. What is the oxygen stoichiometry of BT and KN? It is well known that the complex 3d-metal oxides easily allow the oxygen excess and/or deficit. It can be result of the changes of the electrical properties. Did you estimate oxygen stoichiometry?
7. Conclusion seems poor. Please add most important results and their brief explanation. And check indexes and correct physical values (please replace Jmkg-1K-1V-1 by correct J·m·kg-1·K-1·V-1).
8. There are some typos and grammatical errors in the text. Please revise this.
My decision is major revision. But I impressed by the paper. I feel that after brutal revision it can be accepted.
Reviewer 4 Report
The manuscript by Zhang et al reports on the preparation of electrocaloric material with enhanced properties using a bimodal-structured ceramic. The work is interesting and worth of publication but some issues should be clarified. The most important one is related with the effect of the bimodal distribution on the properties. Thus, in the final section of the manuscript (line 405), it is claimed that “However, in this study, the unimodal structure KN-BT(9/1) using the conventional ceramic processing and sol-gel technique cannot be evaluated the ECEs due to the occurrence of the strong Joule heating. Apparently, the novel bimodal structure KN-BT(9/1) 407 bulk ceramics can overcome the shortage of the unimodal structure by the IAGG method” This is unclear to me. Why the bimodal grain structure does not suffer from Joule heating unlike unimodal structures? Did you get similar densities? Is it related with thermal conductivity? Electrical conductivity? This is a quite important issue that should be clarified.
Other minor issues:
Line 52, where it says “near RT, undera relatively low electric field” IT should say “near RT, under a relatively low electric field”
Line 59 and 60 where it says “in the recent decades, re-59 searchreports” it should say “in the recent decades, re-59 search reports”
Please provide further details about milling procedure (number and diameter of balls, ratio balls/sample, dispersant?, rate, etc.
In experimental section (line 181), section2 .3.4. The setup measurement of electrocaloric effect, it is said that “To directly measure the 182 ECE signals, a pulsed electric field was applied to the sample with a thermocouple (5 pre-183 cision fine wire thermocouples, Omega Engineering, Inc. USA)”. Do you really use 5 thermocouples? In the diagram (figure 1) I can only see one. Could you be more specific about the measurement?
Round 2
Reviewer 2 Report
The authors performed just minor revision of their manuscript that it evidently not enough for the manuscript publication. As a result, a majority of the previous report comments are still valid as following.
Revising the title and making the text more accurate, the authors still have not clarified the meaning of abbreviations like TEM and SAED and symbols like εr and P. Moreover, the authors insist on incorrect calling of the relative dielectric permittivity as dielectric constant that results in odd phrases like “dielectric constant (??,max) increases”, while a constant has to be constant. In addition, estimative words like “good” or “excellent”, which should be avoided in scientific manuscripts, since they depend on point of view, are also kept as well as grammar mistake like “grains exhibits”. Furthermore, unifying the unit style across the manuscript text, the authors have not done that inside of Figs. 6 and 7, keeping as well inconsistent writing “over -20 ‒ 500 °C, at a rate of 1 K/min”.
Then, insisting on showing microstructure of ceramics sintered at 1000 C and 1050 C in the manuscript body, the authors still have not provided density values for both these ceramics as well as conventionally prepared and sol-gel derived ceramics. Moreover, the manuscript still does not specify whether EC effect for 1000 C or for 1050 C ceramics was studied and shown in Fig. 7 as well as there are still no sintering temperature indications in captions of Figs. 5 and 6.
At the same time, there are still many missing details, including the key ones.
Indicating the transition temperatures for KN, those for BT are still absent.
The key detail of the filler content is for some reason presented just in supporting information and again with mistake as “In the present study, ???????? was confined to 10 wt% by the IAGG method”. The requested particle size distribution is not presented at all, while powder TEM images revealing “sizes of about 200‒500 nm” are still in contradiction with the authors’ calling filler powders as “micron-sized” ones.
Request to present rotation rates used at the ball milling step as well as the pot and ball materials is also fully ignored, while the sintering cooling rate and purity of commercial nano-sized BT powder is still not indicated.
XRD range and rate as well as SEM, EDS, TEM acceleration voltages are still not indicated as well.
The information on power supply used for EC measurement and thermal stages or controllers used for temperature dependent dielectric and ferroelectric measurements are also not provided in the manuscript.
The error of heat capacitance and EC measurement has still to be indicated in the manuscript as well as immersion liquid used for density measurements.
(200) XRD peak is shown in supplementary information for all the powders but matrix KN-BT powder. Why? Should one conclude from the provided data that filler KN-BT powder has orthorhombic structure, while BT powder is of cubic structure? Moreover, the lattice parameter values still have to be deduced from XRD with particular need for presentation of relative ceramic density before the discussion of electrical properties.
Concerning the EDS analysis results, instead of showing the requested error values the authors have just moved them to supplementary information. The correspondence of EDS peak slightly below 2 keV to Si contamination is still not checked.
Requested discussion in detail on HRTEM and SAED results, clearly presenting the reasons to exclude O-phase and C-phase, is also not added yet.
Showing the ionic size values, the authors still have not provided a reference or a method, which they used to determine them. Then the authors still have not proven or explained in detail why “When Ti4+ was substituted by Nb5+, the shift (Δ?) of the tetravalent cation from the octahedron center is limited”.
The authors still have not plotted loss tangent in Fig. 5a in log scale, thus showing just that it can be higher than 1 and hence not useful for practical applications but not showing how low it can be.
It is still unclear what the authors mean by “the diffused phase transition can be ascribed to the freezing of the displacive-type phase transition”. How the phase transition can be frozen and that freezing can be another phase transition?
Concerning the contradictions, the authors writing “it is obvious that the temperature (??) of the maximal dielectric constant (??,max) increases with increasing frequency” still contradicts to corresponding Fig. 5a showing opposite variation.
Then, the authors wrote that “the curve of polarization versus electric field at RT is shown as the inset in Fig. 6 (b)”, whereas such curves are shown in Fig. 6 itself, but inset shows remanent polarisation variation. Moreover, based on Fig. 6 the authors claim “the high performance of ferroelectricity” or “high ferroelectricity” in their ceramics. However, the high performance of ferroelectricity is known to be reflected by high saturation polarisation and quadratic loop shape. Where they are in this work?
Further, in their reply the authors wrote that they calculated ΔS as c Δ?/?, the manuscript still contains incorrect equation ΔS = c ρ?Δ?/?, which is still shown by the authors without any reference, without numbering and most important without any relation between ΔT and ΔS values reported in this work.
Finally, Table 1 still shows wrong ΔS/ΔE value of 0.54 instead of 0.93 as the one from Adv. Funct. Mater. 2014, 24, 1300, while the value of 2.3 attributed by the authors to AIP Adv. 2012, 2, 022162 in not mentioned in this cited work at all.
Thus, the authors did not strive to make their manuscript to be useful for readers. As a result, the manuscript cannot be recommended for publication.
Reviewer 3 Report
Accept as is
Author Response
We deeply appreciate the Reviewer's work on reviewing this manuscript.
Round 3
Reviewer 2 Report
Submitted again manuscript confirms that its level does not correspond to that of Nanomaterials, since the authors addressed minor comments ignoring the major ones.
Being awared from the first report on the “particular need for presentation of relative ceramic density before the discussion of its electrical properties”, the authors decided to “delete the contents related to the density in the new updated version.” The cooling rate after sintering that can be also important is not provided as well. Thus, the authors just speculate trying to explain the electrical response.
Being asked from the first report to present powder particle size distributions, the authors keep ignoring this request. Instead they just have shown that they are lost in terms, insisting on calling submicron-sized powders as micron-sized ones.
The authors reject as well to perform Rietveld analysis of their XRD profiles, still hiding a profile for matrix KN-BT powder. The question “Should one conclude from the provided data that filler KN-BT powder has orthorhombic structure, while BT powder is of cubic structure?” is not answered as well.
TEM analysis keeps also to be very incomplete resulting in again speculative conclusion “that the bimodal structured KN-BT(9/1) bulk ceramics have a typical T-phase as mentioned above”.
EDS elemental analysis is again shown without errors and without fit including Si that has to be evidently involved after tens of hours of milling in agate milling media at as high rate as 200 rpm.
Finally, the authors wrote that they got RT dielectric loss as high as 0.067 at 1 kHz and remnant polarisation as low as 0.675 μC·cm-2, but keep concluding that “our bimodal structured bulk ceramic displays high ferroelectricity with low dielectric loss”.
In addition, it is again not reported what devices were used to record the loops and to control temperature “over -20 ‒ 500 °C, at a rate of 1 K·min-1” during temperature-dependent dielectric measurements.
Symbols like εr, εm, Tm and P are again not defined in the text.
Moreover, the authors have shown that they do not understand a difference between the log values and log scale that is needed when a parameter varies by orders of magnitude.
As a result, this manuscript reports results and speculative discussion on Si contaminated ceramics with not reported density, initial powder particle size distribution and lattice parameters. Therefore, it does not correspond to the level of Nanomaterials and cannot be recommended for publication.
